# Publication rates from biomedical and behavioral and social science R01s funded by the National Institutes of Health

**William T. Riley** [1]*, **Katrina Bibb** [2], **Sara Hargrave** [1], **Paula Fearon** [2]

**1** Office of Behavioral and Social Sciences Research, National Institutes of Health, Bethesda, MD, United States of America, **2** Lexical Intelligence, Rockville, MD, United States of America

* wiriley@mail.nih.gov

**Data Availability Statement:** The data from which these analyses were performed is the NIH IMPAC 2 database. Due to the nature of the data stored (e.g.,

## Abstract

Prior research has shown a serious lack of research transparency resulting from the failure to publish study results in a timely manner. The National Institutes of Health (NIH) has increased its use of publication rate and time to publication as metrics for grant productivity. In this study, we analyze the publications associated with all R01 and U01 grants funded from 2008 through 2014, providing sufficient time for these grants to publish their findings, and identify predictors of time to publication based on a number of variables, including if a grant was coded as a behavioral and social sciences research (BSSR) grant or not. Overall, 2.4% of the 27,016 R01 and U01 grants did not have a publication associated with the grant within 60 months of the project start date, and this rate of zero publications was higher for BSSR grants (4.6%) than for non-BSSR grants (1.9%). Mean time in months to first publication was 15.2 months, longer for BSSR grants (22.4 months) than non-BSSR grants (13.6 months). Survival curves showed a more rapid reduction of risk to publish from non-BSSR vs BSSR grants. Cox regression models showed that human research (vs. animal, neither, or both) and clinical trials research (vs. not) are the strongest predictors of time to publication and failure to publish, but even after accounting for these and other predictors, BSSR grants continued to show longer times to first publication and greater risk of no publications than non-BSSR grants. These findings indicate that even with liberal criteria for publication (any publication associated with a grant), a small percentage of R01 and U01 grantees fail to publish in a timely manner, and that a number of factors, including human research, clinical trial research, child research, not being an early stage investigator, and conducting behavioral and social sciences research increase the risk of time to first publication.

## Introduction

There has been an increasing emphasis on replication, openness, and transparency across all of the sciences, including in the health sciences. Many aspects of research transparency have been pursued, including study registration [1] and data sharing [2], but study reporting remains a critical component of research transparency [3]. The National Institutes of Health

proprietary information in grant applications, reviews of applications), the database is restricted to NIH staff. However, we have made a deidentified, publicly accessible database available at https://figshare.com/s/ef60aad738fcb5e2e273

**Funding:** The NIH provided support to the NIH authors (WR, SH) in the form of salaries for the authors, and the Lexical Intelligence authors (KB, PF) were contracted by the NIH to perform an independent analysis of R01/U01 grants and their associated publications. The funder (NIH) did not have any additional role in study design, data analysis, decision to publish, or preparation of the manuscript.

**Competing interests:** The authors have no competing interests. The Lexical Intelligence coauthors (KB, PF) were contracted by the NIH to provide independent analyses of the R01/U01 grants and their associated publications. Although these coauthors have a commercial affiliation, neither they nor their company have any commercial or competing interests regarding the research described in this article. This commercial affiliation does not alter our adherence to PLOS ONE policies on sharing data and materials.

(NIH) clinical trials policies require registration and reporting of results from all experimental studies involving humans to encourage greater research transparency and minimize publication bias [4].

One impetus for these policies was research showing that some studies fail to result in published results. Lack of timely publication and publication bias, particularly from industry-supported trials, have been well-documented [5–7]. Among clinical trials funded by the NIH and registered in ClinicalTrials.gov, only 46% were published within 30 months of trial completion, and a third remained unpublished after an average of 51 months following trial completion [8]. A subsequent analysis of clinical trials funded by the National Heart, Lung, and Blood Institute found that only 64% of these trials had published their primary results 30 months after completion of the trial [9]. Even among larger clinical trials (500 or more participants), 29% remain unpublished, either in the literature or on ClinicalTrials.gov, within an average of 60 months since trial completion [10]. This failure to publish in a timely manner is not unique to clinical trials. Among observational studies evaluating the safety of interventions that were registered in ClinicalTrials.gov, only 39% had published results within 30 months of study completion [11].

Although there is considerable literature on publication bias in behavioral and social sciences research [e.g., 12], we were unable to identify any studies in the literature that specifically evaluated the timeliness of publications in behavioral and social science studies. Some of the clinical trial publication rates reported above include behavioral interventions, and to the degree that behavioral interventions tend to utilize surrogate endpoints (e.g., smoking, weight, blood pressure, cholesterol), these reports suggest that behavioral interventions may be less timely in publishing results than those with clinical endpoints (e.g., morbidity or mortality). While timely and unbiased publication reporting of clinical trials research with potentially immediate impacts on clinical practice is a clear public health need, it is valuable to the scientific enterprise that all types of studies, including basic research, publish in a timely manner.

The NIH has increasingly focused on zero publications as an indicator of productivity, or lack thereof, for the investments that it makes in biomedical and behavioral research. Recently, the NIH began an extensive continuous quality improvement effort of its Center for Scientific Review (CSR) study sections, called ENQUIRE [13]. Among the variables considered in determining the scientific productivity of the grant applications reviewed by study sections is the percentage of funded grants with no publications associated with that grant as per PubMed ID. Further, some preliminary analyses of this criterion suggested that study sections reviewing a higher proportion of behavioral and social sciences research (BSSR) grants had higher rates of zero publication grants. The purpose of this study was to assess the extent of this problem of zero publications, particularly for behavioral and social science grants funded by the NIH, and identify study characteristics associated with R01 grants that fail to publish anything related to that grant within a reasonable period from the project start date.

## Methods

Data used in these analyses were obtained from the NIH Information for Management, Planning, Analysis, and Coordination II (IMPAC II), selecting NIH awarded grants, R01 and U01 type 1s, awarded from 2008 to 2014. Although time to publication is of interest for grant mechanisms other than R01s and U01s, these grant mechanisms are the primary grant mechanisms that would be expected to produce publications, and the timeframe was selected (2008 to 2014) to provide adequate time for grants awarded in 2014 to produce a publication during the typical five-years project period of the typical R01s and U01s.

From the set of awarded grants, we extracted and cleaned the following as potential predictors of time to publication.

- Behavioral and Social Sciences Research (BSSR) vs. not: RCDC coding [14] of BSSR, all other grants not BSSR. This RCDC coding is based on the NIH definition of BSSR—the systematic study of behavioral and social phenomena relevant to health.

- Basic vs. Applied BSSR: RCDC coding of basic BSSR (bBSSR), all other BSSR not coded as bBSSR coded as applied. Neither are all non-BSSR grants.

- Human vs. animal subjects: Determined by IMPAC II flag; coded as human, animal, both, or neither.

- Clinical Trial vs. not: Determined from clinical trial flag as checked on the grant application (Note that these grants predate the current NIH clinical trials definition and policy; therefore, this applicant determined flag likely indicates the traditional efficacy/effectiveness clinical trial, not the broader definition that includes any experimental manipulation.

- Child vs. adult subjects: Determined from study code 2A - Children only, scientifically acceptable, vs study code 3A –No children included, scientifically acceptable. (Note that NIH defined children as those ages $< 21$ years during the time period accessed). Grants were coded as child, adult, or neither.

- Early Stage Investigator (ESI) vs. not: For PI with type 1 grants awarded from 2010–2014, the eRA Commons ESI flag was used. For years 2008 and 2009, ESI status was manually coded based on a) no prior substantial award and b) within 10 years of their terminal degree.

- Time from terminal degree: Calculated in years and equal to fiscal year of grant award minus contact PI's latest degree year. This variable was dichotomized based on median split as $< 17$ years vs. $\geq 17$ years.

- Multiple PI vs. not: Determined if grant award had more than one PI listed as an applicant or not.

- Highest degree received: Obtained for the contact PI and merged into the following categories: Masters, Medical, Doctorate, or Other.

- Carnegie Classification: Institutions receiving grant award were classified as per Carnegie classification of institutions of higher learning, extracted from 2015 definitions and updated in 2018 Public File [15] and merged into the following categories: R1 –Highest research activity doctorate; R2 –Higher research activity doctorate; M1, M2, M3 –moderate research doctoral/masters university or other academia; and special focus—medical schools and centers and other research centers (non-Carnegie medical centers and hospitals included in the special focus category).

Dependent variables were the time to publication and the proportion of grants with zero publications during the 60 months from project start date. PubMed was the source for publications and includes all MEDLINE journal articles plus non-MEDLINE journal articles deposited in PubMed Central, the repository that all NIH research funded by the NIH is required to be deposited. Project start dates and publication dates were exported, and the time calculation determined by subtracting project start date from the publication date of the earliest publication linked to that grant (days). Data removed for analysis included Pubmed IDs (PMIDs) with negative time to first publication values (coded as n/a, n = 15,983 from a total of 456,401 total publications) and those with no publications (coded as n/p, n = 655). A ratio of 30.44:1

was used to convert days to months. Grants were counted as having zero publications if no publications linked to that grant award were identified as being published during the 60 months from the project start date, the typical duration of an R01 or U01 project period.

Survival analysis was carried out in RStudio version 3.6.0 using survival and survminer packages primarily. Kaplan-Meier plots were created to visualize survival curves while log-rank tests were used to compare the survival curves of different groups. Cox hazard regression was executed to describe the effect of predictor variables on time to publication. P-values were computed, and $p < .05$ was considered statistically significant although the size of the sample resulted in small absolute differences being statistically significant. A deidentified, public data-set is available at https://figshare.com/s/ef60aad738fcb5e2e273 for those who wish to replicate findings or conduct additional analyses.

## Results

Of the 27,016 R01 and U01 grants awarded by NIH from 2008 to 2014, 655 grants (2.4%) had zero publications linked to these grants in the 60 months since the project start date. The mean number of publications per grant was 17, and the distribution was positively skewed with a range of 0 to 436, nearly all (99.8%) within a range of 0 to 150. The mean time to first publication was 15.22 months with a range of 1 to 128.

The mean time to first publication for non-BSSR grants was 13.6 months (SD = 13.76), and the mean time to first publication of BSSR grants was 22.44 months (SD = 19.52). Within BSSR, basic BSSR time to first publication was 19.51 months (SD = 17.57) and for applied BSSR, 23.85 months (SD = 20.24). For non-BSSR grants, 421 of 21986 grants (1.9%) had zero publications whereas for BSSR grants, 234 of 5030 (4.6%) had zero publications in the 60 months from the project start date.

Since the proportion of human subjects research (75% for BSSR, 32% for not BSSR) and of clinical trials research (28% BSSR, 7% not BSSR) may partly explain the differences between non-BSSR and BSSR time to publication, Table 1 shows a breakdown of mean time in months to first publication for non-BSSR vs. BSSR (and by basic vs. applied BSSR) by clinical trial and human research code. As shown below, time to publication for clinical trials grants is 7 to 8 months longer on average than for non-clinical trial grants, and BSSR clinical trials, regardless of basic vs. applied subtype, have longer times to publication than non-BSSR clinical trials. Time to publication for human research is also longer, by on average about 7 months, than for other types of research (animal, neither, both). BSSR grants involving research with humans have about a 6-month longer time to first publication than non-BSSR grants.

Kaplan Meier (KM) survival curves for each of the categorical predictors were computed. Fig 1 shows the KM plot for not BSSR (red) and BSSR (green). As shown in the figure, there is a significantly higher probability that BSSR grants will be at risk, especially in the first one to two years of the project, of not publishing by the end of the 60 month period compared to non-BSSR grants, although this risk differential decreases by the 60 month point.

**Table 1. Time to first publication (months) by BSSR or not and by CT and human subject codes.**

| Variable | Strata | Not BSSR | BSSR | Basic BSSR | Applied BSSR |
|---|---|---|---|---|---|
| Clinical Trial | Not CT | 13.1 | 20.2 | 18.9 | 21.0 |
| | CT | 20.8 | 28.4 | 25.9 | 28.7 |
| Human | Neither | 12.4 | 18.7 | 19.3 | 18.3 |
| | Animal | 12.0 | 13.3 | 12.9 | 13.4 |
| | Human | 19.2 | 25.2 | 23.3 | 26.1 |
| | Both | 11.0 | 15.4 | 11.5 | 17.3 |

KM plots for the other categorical covariates were computed. Fig 2 shows the KM plot for human vs. animal research which evidenced the largest survival curve difference for risk of not publishing. As shown in Fig 2, grants coded as human research; compared to animal, neither, or both; had substantially greater risk of not publishing throughout the 60 month period from time of award, with the largest differences in risk occurring in the 12 to 48 month time period, but remaining substantially more at risk even at the 60 month mark. KM plots for other predictor variables showed greater risk of not publishing for applied vs. basic BSSR, for clinical trials vs. not, and for child vs. adult.

Among PI and institutional predictors, while being an ESI or being less than 17 years from the terminal degree significantly reduced the risk of not publishing, the absolute differences in the curves were small. The ESI KM plot is shown in Fig 3.

To assess the relative predictive value of these variables controlling for the others, we performed a Cox regression analysis. Table 2 shows the hazard ratios of the full Cox model compared to reference. All of the predictors included in the model were significant with the exception of time to terminal degree which is highly related to ESI status. Accounting for the effects attributable to other predictors, human research has among the lowest hazard ratio for risk of not publishing (.65) compared to the neither reference category (animal and both slightly higher). Even accounting for human vs. animal participants and clinical trial vs. not, both basic and applied BSSR had a lower risk of publishing (.78 and .77 respectively) relative to non-BSSR grants. Being an ESI, having a medical or doctorate degree, and submitting a multiple PI grant were all predictive factors that increased the likelihood of publishing in a timely manner.

A comparison of simple and multiple Cox models along with separate Schoenfeld residual p values based on unstratified covariates revealed that the hazards of the strata were not proportional to one another. Therefore, stratified models were computed (e.g., Cox regressions for BSSR only) and a dichotomous time to terminal degree was computed. These stratified models revealed some differences to the full model in the absolute hazard ratio, but the pattern of predictors across most stratified models were consistent with the full model. Table 3 below shows the Cox model stratified by BSSR or not-BSSR. For both groups, human research substantially increases the hazard of not publishing, as does conducting a clinical trial. Being an ESI or having less than 17 years since terminal degree reduces the hazard of not publishing.

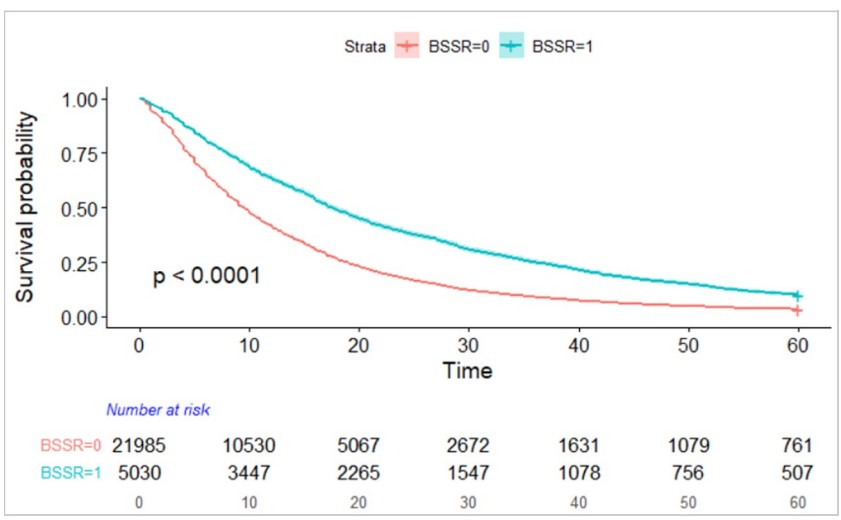

**Fig 1. KM survival curve of time to first publication for BSSR and non-BSSR grants.**

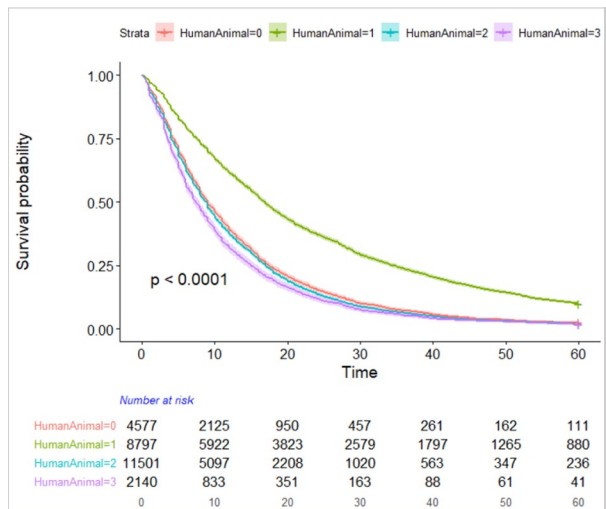

**Fig 2. KM survival curve of time to first publication for human and animal research codes.**

## Discussion

The results of these data from over 27,000 NIH R01 and U01 grants awarded between 2008 and 2014 show that 2.4% of these awards had zero publications associated with the grant award in the 60 months since the start of the project period. This rate of zero publications is much less than found in prior studies. Most prior studies, however, focused on publication of the primary outcome results from clinical trials. In contrast, this study included all NIH R01 and U01 grants, only a small proportion of which were conducting clinical trials, and used a much more liberal outcome criteria, any PubMed ID publication associated with the grant, not the primary outcome results from the project. This approach may underestimate the true rate of zero publications resulting from a specific grant since investigators can associate any grant to a publication, regardless of how tangential that publication may be to that grant. Consistent with this possibility that publications are associated with grants that may have not

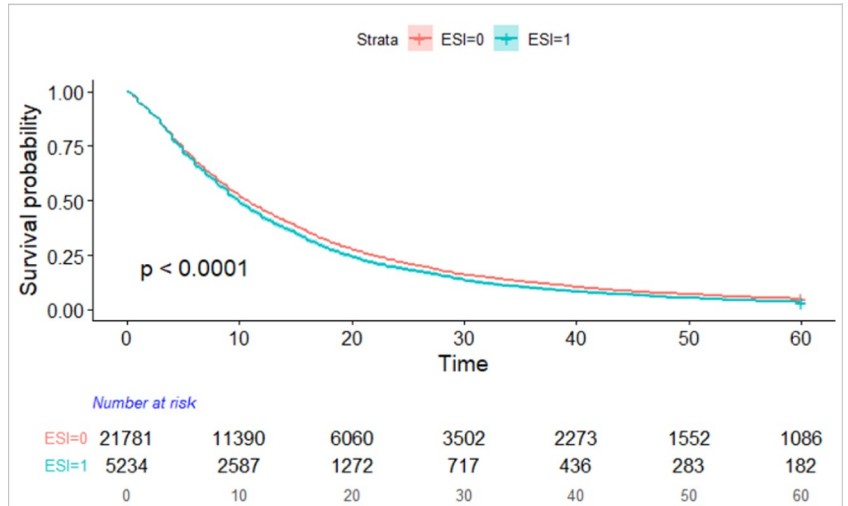

**Fig 3. KM survival curve of time to first publication for ESI (1) vs. not-ESI (0) investigators.**

**Table 2. Covariate and hazard ratios for full Cox model.**

|  | Strata | Exp(β) | 95% CI± | p-value | β coef | SE |
|---|---|---|---|---|---|---|
| **BasicApplied** | 1 = Basic | 0.78 | 0.74–0.83 | < 2e-16 *** | -0.24 | 0.03 |
|  | 2 = Applied | 0.77 | 0.73–0.80 | < 2e-16 *** | -0.27 | 0.02 |
| **HumanAnimal** | 1 = Human | 0.65 | 0.62–0.68 | < 2e-16 *** | -0.43 | 0.02 |
|  | 2 = Animal | 1.05 | 1.02–1.10 | 0.00159 ** | 0.06 | 0.02 |
|  | 3 = Both | 1.15 | 1.08–1.21 | 1.12e-06 *** | 0.14 | 0.03 |
| **CT** | 1 = yes | 1.05 | 0.76–0.83 | < 2e-16 *** | -0.23 | 0.02 |
| **ChildAdult** | 1 = Child | 0.93 | 0.86–1.00 | 0.06 | -0.08 | 0.04 |
|  | 2 = Adult | 1.06 | 1.01–1.11 | 0.01162 * | 0.06 | 0.02 |
| **TFTD** | n/a | 0.99 | 0.99–0.99 | < 2e-16 *** | -0.01 | 0.00 |
| **ESI** | 1 = yes | 1.05 | 1.01–1.09 | 0.00805 ** | 0.05 | 0.02 |
| **MultiplePIs** | 1 = yes | 1.14 | 1.10–1.18 | 3.44e-13 *** | 0.13 | 0.02 |
| **Degree** | 1 = Medical | 1.18 | 1.10–1.25 | 3.41e-07 *** | 0.16 | 0.03 |
|  | 2 = Doctorate | 1.12 | 1.06–1.19 | 4.77e-05 *** | 0.12 | 0.03 |
|  | 3 = Other | 1.02 | 0.86–1.20 | 0.83 | 0.12 | 0.08 |
| **Carnegie** | 1 = R2 | 0.98 | 0.93–1.03 | 0.43 | -0.02 | 0.03 |
|  | 2 = Other academia | 0.96 | 0.90–1.03 | 0.29 | -0.04 | 0.04 |
|  | 3 = Medical school/center | 0.97 | 0.94–1.00 | 0.07 | -0.03 | 0.02 |
|  | 4 = Research/Other | 0.73 | 0.70–0.76 | < 2e-16 *** | -0.31 | 0.02 |

BSSR removed above from full model due to its inclusion in BasicApplied neither level; Exp(β): Hazard Ratio (HR) per predictor after accounting for all other predictors in model CI: Confidence Interval, ±: for Exp(β), Significance codes:

*** at p = 0.001,

** at p = 0.01,

* at p = 0.05 β: regression coefficient, SE: Standard Error.

contributed in any substantive way to the publication, we identified during the data cleaning process nearly 16,000 publications (about 3.5% of total publications) for which the publication preceded the project start date and were excluded from analyses because our results otherwise would have negative values for time to publication from the grant start date. Given that linking a grant number to a publication is all that is required to meet the criterion of a publication resulting from a grant, the fact that even 2.4% of R01 and U01 grants (655 NIH funded R01s or U01s) failed to be associated with even one publication in the 60 months following the project start date is potentially problematic for research transparency.

The mean length of time to the first publication was approximately 15 months from the project start date, and this time to first publication varies substantially based on the type of research being conducted. Clinical trials add, on average, approximately 9 months to the time to first publication, and research with humans adds, on average, approximately 7 months to the time to first publication. These are not unexpected findings. Human subjects research requires Institutional Review Board (IRB) approvals and informed consent procedures. Long-term longitudinal studies take more time to conduct than cross-sectional studies. Recruitment and retention of human subjects in research studies is a significant challenge that often delays study timelines.

One purpose of this study was to examine further some preliminary NIH findings suggesting that behavioral and social sciences research grants and study sections may not be as productive as non-BSSR or biomedical research study sections. Evaluations of study sections using the Relative Citation Ratio (RCR) [16] had shown that the RCRs from some BSSR-focused study sections may not be as high as for some more biomedically focused

**Table 3. Full Cox regression models stratified by BSSR or not-BSSR.**

| Covariate | Strata | BSSR only n = 4927 | | | Non-BSSR n = 21566 | | |
|---|---|---|---|---|---|---|---|
| | | Exp(β) | 95% CI | p-value | Exp(β) | 95% CI | p-value |
| BasicApplied^ | 2 = Applied | 0.99 | 0.93–1.06 | 0.77 | - | - | - |
| HumanAnimal | 1 = Human | 0.73 | 0.64–0.84 | 3.44e-06 *** | 0.66 | 0.63–0.70 | < 2e-16 *** |
| | 2 = Animal | 1.50 | 1.30–1.72 | 1.08e-08 *** | 1.03 | 1.00–1.07 | 0.07 |
| | 3 = Both | 1.40 | 1.04–1.86 | 0.03* | 1.14 | 1.07–1.21 | 6.46e-06 *** |
| CT | 1 = yes | 0.77 | 0.71–0.83 | 5.05e-12 *** | 0.81 | 0.76–0.86 | 5.97e-11 *** |
| ChildAdult | 1 = Child | 1.07 | 0.95–1.20 | 0.26 | 0.83 | 0.75–0.93 | 0.000873 *** |
| | 2 = Adult | 1.06 | 0.97–1.15 | 0.20 | 1.05 | 1.00–1.10 | 0.07 |
| TFTD2 | 1: ≥ 17 years | 0.89 | 0.84–0.96 | 0.003 ** | 0.89 | 0.89–0.92 | 3.68e-12 *** |
| ESI | 1 = yes | 1.08 | 0.99–1.17 | 0.06 | 1.07 | 1.03–1.11 | 0.000714 *** |
| MultiplePIs | 1 = yes | 1.06 | 0.99–1.15 | 0.14 | 1.15 | 1.11–1.20 | 5.76e-13 *** |
| Degree | 1 = Medical | 1.15 | 1.01–1.31 | 0.03 * | 1.16 | 1.08–1.25 | 3.17e-05 *** |
| | 2 = Doctorate | 1.03 | 0.93–1.14 | 0.57 | 1.13 | 1.06–1.22 | 0.000201 *** |
| | 3 = Other | 0.95 | 0.72–1.26 | 0.72 | 1.04 | 0.85–1.28 | 0.70 |
| Carnegie | 1 = R2 | 1.10 | 0.97–1.25 | 0.12 | 0.96 | 0.90–1.01 | 0.12 |
| | 2 = Other academia | 0.99 | 0.82–1.19 | 0.93 | 0.97 | 0.89–1.04 | 0.38 |
| | 3 = Medical school/center | 0.98 | 0.90–1.06 | 0.55 | 0.97 | 0.93–1.00 | 0.06 |
| | 4 = Research/Other | 0.80 | 0.73–0.88 | 9.96e-06 *** | 0.72 | 0.69–0.76 | < 2e-16 *** |

-: not in model, Exp(β): Hazard Ratio (HR) per predictor *after* accounting for all other predictors in model CI: Confidence Interval, Significance codes:

*** at p = 0.001,

** at p = 0.01,

* at p = 0.05 β.

study sections. Although RCR normalizes the citation behavior of a research community, it is a publication-level metric that does not normalize the publication behavior of a research community when aggregated across investigators, study sections, institutes, etc. Therefore, when the CSR Enquire effort began, a simpler index of the proportion of zero publications was considered.

This study shows that, even after accounting for the disproportionate rate of human subjects research and clinical trials among BSSR grants, BSSR grants still have a higher risk of not publishing within a 5 year period from their project start date than their non-BSSR biomedical counterparts. Without controlling for other variables, the average time to first publication is 14 months for a non-BSSR grant and 22 months for a BSSR grant. Within clinical trial grants, BSSR is slower to first publication than non-BSSR, and the same holds true for human subjects research. The survival curves for risk of not publishing drop substantially faster for non-BSSR vs BSSR grants. After accounting for all other variables in the model, the risk of not publishing is 22 to 23 percent less for BSSR compared to non-BSSR research. It is possible that BSSR research takes inherently longer to conduct, even after accounting for other variables, and without publication data beyond 60 months, we do not know if, with more time, these non-BSSR and BSSR curves continue out until near zero risk of not publishing. However, the data from this study and from others [17] suggest that likelihood of publishing decreases over time.

Although the focus of this study was primarily on how the types of studies proposed in grants (BSSR or not, clinical trial or not, human research or not, child or not) affect the time to publication and the risk of not publishing, we also were able to assess certain awardee (both PI and institution) characteristics associated with the risk of not publishing. Although there was little differentiation among institutions based on Carnegie classification, grants to "research

other" institutions had a higher risk of not publishing than the R1 and R2 research institutions. "Research other" is a heterogenous group that includes medical centers unaffiliated with academic institutions, private research institutions, and institutions without a Carnegie category. These institutions may have less resources to assist investigators in conducting studies and publishing in a timely manner.

Although being an ESI or being less than 17 years from terminal degree was not associated with a substantial reduction in risk of not publishing, it was a significant reduction compared to those who were not ESIs or those with 17 or more years since terminal degree. These findings add further support to the NIH's Next Generation Researchers Initiative [18]. Being on a multiple PI grant also reduced the risk of not publishing. This may simply be the result of having more PI leadership for publishing results, and/or it may be related to the transdisciplinary nature of some multi-PI grants. Prior research has shown that while slower to publish initially, transdisciplinary teams eventually publish more than non-transdisciplinary projects [19].

This study has a number of limitations to consider when interpreting these results. As noted previously, the association of a PMID to a grant is a liberal criterion for a publication that resulted from a research grant. Given the large sample size, manually identifying the primary or clearly relevant publications from a given grant was not feasible. We identified publications for up to 60 months from the project start date, and for some larger longitudinal studies, the first publication may occur beyond that time period. The predictors were selected based on their availability in the NIH IMPACII grant database; other predictors of risk of not publishing may be important, but we were not able to consider them given the limited number of relevant variables in this database. The criteria for categorizing these predictors also were based on the tools available (e.g., RCDC for BSSR, investigator checking "clinical trial" on an application), which may produce categorization errors.

Within these limitations, the findings from this study show that while most NIH R01 and U01 grants publish at least one publication within 5 years of the project start date, a small percentage do not. There are legitimate reasons for no publications to result from a research grant (e.g., unfeasible to conduct, design flaws that limit internal validity and reproducibility, negative results that are difficult to publish), but for R01 and U01 grants that require sufficient preliminary data to demonstrate that the proposed research is feasible and is likely to result in meaningful results that will advance the science, it is also reasonable for taxpayers to question their investment in research grants that produce no publications.

Based on these analyses, there appears to be a higher risk of not publishing in a timely manner if the research involves humans, clinical trials, children, or behavioral and social sciences research. Since these types of research increase the risk of not publishing in a timely manner, increased monitoring of progress, such as per NIH's clinical trials monitoring policies, appears appropriate. As noted previously, the Center for Scientific Review (CSR) considers zero publication rates from grants as criteria for its ENQUIRE process of evaluating study sections. The publication productivity of grantees is typically considered in Type 2 applications, and there are examples such as in the National Institute of Child Health and Human Development (NICHD) Neonatal Research Network in which investigators were restricted from proposing new grants until the manuscripts from their prior work were completed [20]. Increased weight in review and in funding decisions on the quality and timeliness of publications from prior grant awards of the research team has the potential to increase the productivity and transparency of research funded by the NIH. Further research on the factors associated with the risk of not publishing could identify grants that require additional monitoring, support, and incentives to publish so that the taxpayer funding of NIH-supported research results in transparent findings that advance the science.

## Author Contributions

**Conceptualization:** William T. Riley, Sara Hargrave.

**Data curation:** Katrina Bibb, Sara Hargrave.

**Formal analysis:** Katrina Bibb, Paula Fearon.

**Investigation:** Sara Hargrave.

**Methodology:** William T. Riley, Katrina Bibb.

**Project administration:** William T. Riley, Paula Fearon.

**Resources:** William T. Riley, Sara Hargrave.

**Supervision:** William T. Riley, Paula Fearon.

**Validation:** Paula Fearon.

**Writing – original draft:** William T. Riley, Katrina Bibb, Sara Hargrave, Paula Fearon.

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
