## [Decision Letter · Decision Letter 0]

11 Sep 2020

PONE-D-20-26044

Rates of Zero Publications from Biomedical and Behavioral and Social Science R01s funded

by the National Institutes of Health

PLOS ONE

Dear Dr. Riley,

Thank you for submitting your manuscript to PLOS ONE. After careful consideration, we feel that it has merit but does not fully meet PLOS ONE’s publication criteria as it currently stands. Therefore, we invite you to submit a revised version of the manuscript that addresses the points raised by an  expert reviewer and additional editor comments during the review process.

We look forward to receiving your revised manuscript.

Kind regards,

Dr. Sakamuri V. Reddy

Academic Editor

PLOS ONE

Journal Requirements:

"The authors received no specific funding for this work."

We note that one or more of the authors are employed by a commercial company: "Lexical Intelligence"

Additional Editor Comments (if provided):

The manuscript focusses on NIH supported R01 and U01 grants and publication lag. This study predicts the reasons for publication lag in type of research and investigator status. However, to further improve the manuscript, please clarify the Methods section at the end for Statistical analysis of data and significance considered. Although, the authors have given time for first publication, they should clarify discussion for variability reasons in publication time between clinical trial grants vs R01 grants or human subjects vs animal subjects involvement in the study. Also, publication time variability for early stage vs established investigators. They may discuss the difficulty to identify or analyze abstract of the studies presented in a scientific society meeting annually.

Reviewers' comments:

Reviewer's Responses to Questions

**Comments to the Author**

1. Is the manuscript technically sound, and do the data support the conclusions?

Reviewer #1: Partly

2. Has the statistical analysis been performed appropriately and rigorously? 

Reviewer #1: No

3. Have the authors made all data underlying the findings in their manuscript fully available?

Reviewer #1: No

4. Is the manuscript presented in an intelligible fashion and written in standard English?

Reviewer #1: Yes

5. Review Comments to the Author

Reviewer #1: Reviewer’s Preface

There may be a dearth of literature evaluating the timeliness of publications in behavioral and social science studies as the authors point out, but there are many good peer-reviewed studies on publication rates for human clinical studies. Of the latter, publication rates are most often based upon the study completion date, and publication in an indexed, peer-reviewed journal. Since this manuscript is likely to be read by healthcare professionals familiar with previous methodologies for publication rate studies, the authors should explain why theirs differ. The following review offers some specific observations and suggestions.

Methods and Results:

BSSR vs. non-BSSR Grants: The authors assume the reader is familiar with BSSR grants. Could the authors explain the difference between BSSR and non-BSSR grants in a few sentences?

Term of the Study: Many previous publication rate reports are based on post-study completion, making a direct comparison to this study difficult. Miller (2015) evaluated publication rates based upon Section 801 of the Food and Drug Administration Amendments Act of 2007 (FDAAA) requiring posting of study results no later than one year after the date of completion or early termination. Prenner (2011), Chen (2016) and Archer (2016) evaluated publication rates 24 months post study completion. Gordon (2014) chose to evaluate publication rates 30 months post study completion. Post-study completion takes into account the duration of the study. Post-grant award does not. Can the authors explain their rationale for using “60 months from grant award” as the term of this study?

Publication Rates: A zero publication rate of 2.4% seems low and may be the result of the type of study (BSSR), term of the study (60 months from project implementation) and the author’s liberal criteria for publication. Gordon (2014) reported a publication rate of 64% 30 months post-study, Archer (2016) 64% after 2 years, Miller (2015) 67% after 1 year, Prenner (2011) 54% after 2 years, Chen (2016) 36% after 24 months. Can the authors provide further clarification for the difference in their publication rates?

Definition of Publication: Authors refer to their “liberal criteria for publication” without clearly defining what constitutes a publication. Were these citable, peer-reviewed publications that appeared in indexed journals? Were they non-peer-reviewed communication activities such as a posting on a website, or on social media?

Publications per Grant: The range of publications per grant was 0 to 436 (pg. 7). The latter number is large and may confound the mean. Was the Standard Deviation calculated? Could a handful of grants skew the data? If so, should they should be identified and removed from the calculation to provide an accurate picture of the range of publications per grant?

Publications Identified: The authors note in two locations that 16,000 pre-project publications were excluded from the analysis (pp 7, 14), but the total number of publications that met their criteria for inclusion does not appear in the manuscript. We know that 27,016 grants were included (pp 2, 7). We know the average number of publications per grant was 17 (pg. 7). Is the reader to assume that 459,272 publications were identified (17 x 27,016)?

Size of Grant: Did the size of the study grant have any bearing on the rate of publication?

Human Studies: Studies on human subjects require review by an Institutional Review Board and often the informed consent of the subject as well. Could this account for the 7-8 month delay in publication (pg. 8)?

Publishing Incentives: Could the incentives for publishing at “research other” institutions (pg 16) relate to the fact that smaller institutions do not have the resources of an academic center (eg. post-docs, students, and professional medical writers who can assist with reconciling case report forms, collating data, preparing tables, and drafting and submitting study manuscripts)?

Study Limitations: Mostly addressed above, with one addition. Does the sheer size of the data set and the comparison of disparate subgroups preclude the nuanced analysis necessary to recommend and implement specific policy changes?

Discussion:

Zero Risk of Not Publishing: Is the achievement of zero rates of non-publication (pg. 15) a practical goal? Many studies fail to achieve their objectives. Knowing how many studies failed, and the reasons behind the study’s failure is useful for future researchers as they design their protocols. However, there is little interest on the part of investigators and journal editors to spend time on negative or irreproducible results. Could this be a reason why some grants will never have an associated publication? Do the authors know how many grants had negative or irreproducible results?

Additional Recommendations Needed: The authors recommend further support for the NIH’s Next Generation Researchers Initiative (pg. 16). Could additional successful examples be included such as the National Institute of Child Health and Human Development Neonatal Research Network (Archer 2016) that restricted investigators with unfinished manuscripts from proposing new grant requests? Would a policy similar to that of the NRN help to improve BSSR publication rates by holding investigators accountable for publication?

Some Editorial Comments

Manuscript Title: Would on-line searchability improve if the title was simplified as follows: “Publication Rates from Biomedical and Social Science R01s by the National Institutes of Health.” Will the word “zero” in the title serve to confound a future literature search?

Typo: Pg. 12, second sentence. Should the word be “predictive” instead of “protective”?

Reviewer Citations

Archer SW, Carlo WA, Truog WE, et. al. Improving publication rates in a collaborative clinical trials research network. Eunice Kennedy Shriver National Institute of Child Health and Human Development Neonatal Research Network. Semin Perinatol. 2016 Oct;40(6):410-417.

Chen R, Desai NR, Ross JS, et. al. Publication and reporting of clinical trial results: cross sectional analysis across academic medical centers. BMJ. 2016 Feb 17;352:i637.

Food and Drug Administration Amendments Act of 2007 (FDAAA), Section 801

Gordon D, Taddei-Peters W, Mascette A, et. al. Publication of trials funded by the National Heart, Lung, and Blood Institute. N Engl J Med. 2013 Nov 14;369(20):1926-34.

Miller JE, Korn D, Ross JS. Clinical trial registration, reporting, publication and FDAAA compliance: a cross-sectional analysis and ranking of new drugs approved by the FDA in 2012. BMJ Open. 2015 Nov 12;5(11)

Prenner JL, Driscoll SJ, Fine HF, Salz DA, Roth DB. Publication rates of registered clinical trials in macular degeneration. Retina. 2011 Feb;31(2):401-4.

6. PLOS authors have the option to publish the peer review history of their article (what does this mean?). If published, this will include your full peer review and any attached files.

Reviewer #1: No

---

## [Author Response · Author response to Decision Letter 0]

28 Oct 2020

October 26, 2020

Dr. Sakamuri V. Reddy

Academic Editor

PLOS ONE

Re: PONE-D-20-26044, “Rates of Zero Publications from Biomedical and Behavioral and Social Science R01s funded by the National Institutes of Health”

PLOS ONE

Dr. Reddy,

Thank you and the reviewer for the comments on this manuscript. We have addressed each point raised in this response and submitted a marked-up manuscript based on these review points and our responses as well as a clean version of the revised manuscript. 

1. Please ensure that your manuscript meets PLOS ONE's style requirements, including those for file naming. We have rechecked the manuscript to ensure that it adheres to the PLOS ONE style requirements.

2. We note that you have indicated that data from this study are available upon request. PLOS only allows data to be available upon request if there are legal or ethical restrictions on sharing data publicly.

Although the data for this study were extracted from NIH’s IMPACII database which is restricted access to protect grantees and their proprietary information, we have created a deidentified dataset (i.e., no grant number, appl ID, PI name, or PI institution) that can be made available for those who desire to reproduce our results. 

We have created a deidentified dataset that is posted on figshare.com for others to access for replication or further analyses.

"The authors received no specific funding for this work."

We note that one or more of the authors are employed by a commercial company: "Lexical Intelligence"

a) Please provide an amended Funding Statement declaring this commercial affiliation, as well as a statement regarding the Role of Funders, in your study. If the funding organization did not play a role in the study design, data collection and analysis, decision to publish, or preparation of the manuscript and only provided financial support in the form of authors' salaries and/or research materials, please review your statements relating to the author contributions, and ensure you have specifically and accurately indicated the role(s) that these authors had in your study. You can update author roles in the Author Contributions section of the online submission form. 

We have updated Author Contributions to indicate that Lexical Intelligence was contracted by the NIH to perform these analyses. We have amended the funding statement to reflect this. We did amend that statement above slightly since the funder, NIH, was involved in the data collection as part of its administrative responsibilities to monitor grants and their associated publications, but NIH did not have a role in any of the other study roles described above. 

We have added this statement.

Lexical Intelligence was contracted by the NIH to perform these analyses and does not have any conflict of interest regarding the results of the study. Actually, having an independent contractor perform these analyses eliminates any appearance of conflict the NIH might have in publishing results of the productivity of the grants it awards.

We have updated the Competing Interests Statement declaring the commercial affiliation of the co-authors.

We have confirmed that this commercial affiliation does not alter adherence to all PLOS ONE policies.

We have done so in our cover letter. 

Please know it is PLOS ONE policy for corresponding authors to declare, on behalf of all authors, all potential competing interests for the purposes of transparency. PLOS defines a competing interest as anything that interferes with, or could reasonably be perceived as interfering with, the full and objective presentation, peer review, editorial decision-making, or publication of research or non-research articles submitted to one of the journals. Competing interests can be financial or non-financial, professional, or personal. Competing interests can arise in relationship to an organization or another person. Please follow this link to our website for more details on competing interests: http://journals.plos.org/plosone/s/competing-interests. 

We appreciate the competing interest efforts of PLOS ONE, but in this case, the commercial entity has less competing interest in a full and objective presentation of the results than does the NIH which could be perceived as wanting their grant awards to appear productive regarding publication rates, hence why we contracted to have an independent analysis of these data. 

Additional Editor Comments (if provided):

The manuscript focuses on NIH supported R01 and U01 grants and publication lag. This study predicts the reasons for publication lag in type of research and investigator status. However, to further improve the manuscript, please clarify the Methods section at the end for Statistical analysis of data and significance considered. Although, the authors have given time for first publication, they should clarify discussion for variability reasons in publication time between clinical trial grants vs R01 grants or human subjects vs animal subjects involvement in the study. Also, publication time variability for early stage vs established investigators. They may discuss the difficulty to identify or analyze abstract of the studies presented in a scientific society meeting annually. 

We have clarified in the methods section the statistical analyses and significance considered. We also have further clarified in the discussion some of the potential reasons for the differences noted in publication lags between types of grants. 

Reviewers' comments:

Reviewer's Responses to Questions 

Comments to the Author

Reviewer #1: Reviewer’s Preface

There may be a dearth of literature evaluating the timeliness of publications in behavioral and social science studies as the authors point out, but there are many good peer-reviewed studies on publication rates for human clinical studies. Of the latter, publication rates are most often based upon the study completion date, and publication in an indexed, peer-reviewed journal. Since this manuscript is likely to be read by healthcare professionals familiar with previous methodologies for publication rate studies, the authors should explain why theirs differ. The following review offers some specific observations and suggestions.

Methods and Results:

BSSR vs. non-BSSR Grants: The authors assume the reader is familiar with BSSR grants. Could the authors explain the difference between BSSR and non-BSSR grants in a few sentences? 

We have included the NIH definition of BSSR that is the basis for RCDC coding of grants as BSSR. 

Term of the Study: Many previous publication rate reports are based on post-study completion, making a direct comparison to this study difficult. Miller (2015) evaluated publication rates based upon Section 801 of the Food and Drug Administration Amendments Act of 2007 (FDAAA) requiring posting of study results no later than one year after the date of completion or early termination. Prenner (2011), Chen (2016) and Archer (2016) evaluated publication rates 24 months post study completion. Gordon (2014) chose to evaluate publication rates 30 months post study completion. Post-study completion takes into account the duration of the study. Post-grant award does not. Can the authors explain their rationale for using “60 months from grant award” as the term of this study? 

We have explained this further in the revised manuscript. The source of our data is different from those from clinical trials studies in that our source includes all research, not just clinical trials research. As a result, we are only able to specify the project start date, not the study completion date. Therefore, we chose 60 months from grant award to include the full 5-year period of most U01 and R01 awards. We considered a longer time period to account for no cost extensions but the longer the window, the more dated the data become to provide sufficient time for publication. We believe 60 months is a reasonable compromise between providing sufficient time for most grants to publish something during their project period but not so long as to push back further the inclusion criteria for when grants were initially awarded. 

Publication Rates: A zero publication rate of 2.4% seems low and may be the result of the type of study (BSSR), term of the study (60 months from project implementation) and the author’s liberal criteria for publication. Gordon (2014) reported a publication rate of 64% 30 months post-study, Archer (2016) 64% after 2 years, Miller (2015) 67% after 1 year, Prenner (2011) 54% after 2 years, Chen (2016) 36% after 24 months. Can the authors provide further clarification for the difference in their publication rates? 

We discussed this in the original manuscript but have further elaborated on these differences in this revision. Most prior research on publication lags has focused on clinical trials and assessed the time to the publication of the primary results. That more stringent criterion for publication is not possible or appropriate within the dataset we have analyzed. Instead, this data set only allows us to identify any and all publications that are associated with a given grant in PubMed Central. Therefore, any publication associated with a grant, not the primary outcome publication from a clinical trial, is included. Further, as indicated by our results, clinical trials have longer lag times to any publication than do grants that are not clinical trials. 

Definition of Publication: Authors refer to their “liberal criteria for publication” without clearly defining what constitutes a publication. Were these citable, peer-reviewed publications that appeared in indexed journals? Were they non-peer-reviewed communication activities such as a posting on a website, or on social media? 

We have clarified what we mean by “liberal criteria” – any publication associated with a grant in PubMed Central. Most of these publications are peer-reviewed journal publications although PubMed Central does sometimes include proceedings from meetings if published in a journal. It, however, does not include posting to websites, social media, etc. By “liberal” criteria, we mean that all that is required is for an investigator to login to PubMed Central and associate a grant, or multiple grants, with a publication. 

Publications per Grant: The range of publications per grant was 0 to 436 (pg. 7). The latter number is large and may confound the mean. Was the Standard Deviation calculated? Could a handful of grants skew the data? If so, should they be identified and removed from the calculation to provide an accurate picture of the range of publications per grant? 

The reviewer is correct that this distribution is positively skewed, but there are no obvious outliers in this distribution. We have included further information on the distribution of publications per grant in the revised manuscript, but it is important to note that we provide the number of publications per grant in this manuscript only for descriptive purposes. One reason we focused our analyses on time to publication and presence or absence of any publication is because the number of publications is so positively skewed.

Publications Identified: The authors note in two locations that 16,000 pre-project publications were excluded from the analysis (pp 7, 14), but the total number of publications that met their criteria for inclusion does not appear in the manuscript. We know that 27,016 grants were included (pp 2, 7). We know the average number of publications per grant was 17 (pg. 7). Is the reader to assume that 459,272 publications were identified (17 x 27,016)? 

We have included the total number of publications associated with grants (456,401) and the number excluded because they were associated before the project start date. 

Size of Grant: Did the size of the study grant have any bearing on the rate of publication? 

We restricted our analyses to R01s and U01s in part to assess the publication rates of grants of comparable funding size. One might expect differences in publication rates for smaller (R03, R21) or larger (P30, P50) grants, but for R01s and U01s, the grant funding sizes are quite similar. 

Human Studies: Studies on human subjects require review by an Institutional Review Board and often the informed consent of the subject as well. Could this account for the 7-8 month delay in publication (pg. 8)? 

This is a good point and we’ve added it to the potential reasons for the delay in conducting and publishing studies involving humans.

Publishing Incentives: Could the incentives for publishing at “research other” institutions (pg 16) relate to the fact that smaller institutions do not have the resources of an academic center (eg. post-docs, students, and professional medical writers who can assist with reconciling case report forms, collating data, preparing tables, and drafting and submitting study manuscripts)? 

We agree that this is a potential cause of the difference, and hence why we considered Carnegie categories as a predictor variable. We have made this point more explicit in the revised manuscript.

Study Limitations: Mostly addressed above, with one addition. Does the sheer size of the data set and the comparison of disparate subgroups preclude the nuanced analysis necessary to recommend and implement specific policy changes? 

More nuanced analyses could reveal potential policy changes that would encourage publication and as a program evaluation effort, NIH regularly assesses publication rates of specific types of grants and considers policy changes, but for the purposes of this manuscript, we wanted to document the overall rates of publication, publication time-lags, and predictors of those lags in the most common form of grant that the NIH awards. By making a deidentified public dataset available, we hope others can pursue more nuanced questions as well. 

Discussion:

Zero Risk of Not Publishing: Is the achievement of zero rates of non-publication (pg. 15) a practical goal? Many studies fail to achieve their objectives. Knowing how many studies failed, and the reasons behind the study’s failure is useful for future researchers as they design their protocols. However, there is little interest on the part of investigators and journal editors to spend time on negative or irreproducible results. Could this be a reason why some grants will never have an associated publication? Do the authors know how many grants had negative or irreproducible results? 

This is an excellent point that we have attempted to make more explicit in the discussion section. We don’t know how many grants had negative or irreproducible results. It is true that some studies turn out to be unfeasible to carry out and others are carried out, produce negative results, and are unable to publish in light of publication bias. We don’t believe the optimal number of grants with no publications should be zero, but given the typical $3 to $4 million investment of a typical R01 and U01, some published contribution from nearly all grants seems reasonable.

Additional Recommendations Needed: The authors recommend further support for the NIH’s Next Generation Researchers Initiative (pg. 16). Could additional successful examples be included such as the National Institute of Child Health and Human Development Neonatal Research Network (Archer 2016) that restricted investigators with unfinished manuscripts from proposing new grant requests? Would a policy similar to that of the NRN help to improve BSSR publication rates by holding investigators accountable for publication? 

This is an excellent point and we have added this example as well as other potential strategies to improve timely publication.

Some Editorial Comments

Manuscript Title: Would on-line searchability improve if the title was simplified as follows: “Publication Rates from Biomedical and Social Science R01s by the National Institutes of Health.” Will the word “zero” in the title serve to confound a future literature search? 

Thank you for this title revision. We have modified the title accordingly. 

Typo: Pg. 12, second sentence. Should the word be “predictive” instead of “protective”? 

Yes, this has been corrected.

Reviewer Citations

Archer SW, Carlo WA, Truog WE, et. al. Improving publication rates in a collaborative clinical trials research network. Eunice Kennedy Shriver National Institute of Child Health and Human Development Neonatal Research Network. Semin Perinatol. 2016 Oct;40(6):410-417.

Chen R, Desai NR, Ross JS, et. al. Publication and reporting of clinical trial results: cross sectional analysis across academic medical centers. BMJ. 2016 Feb 17;352:i637.

Food and Drug Administration Amendments Act of 2007 (FDAAA), Section 801

Gordon D, Taddei-Peters W, Mascette A, et. al. Publication of trials funded by the National Heart, Lung, and Blood Institute. N Engl J Med. 2013 Nov 14;369(20):1926-34.

Miller JE, Korn D, Ross JS. Clinical trial registration, reporting, publication and FDAAA compliance: a cross-sectional analysis and ranking of new drugs approved by the FDA in 2012. BMJ Open. 2015 Nov 12;5(11)

Prenner JL, Driscoll SJ, Fine HF, Salz DA, Roth DB. Publication rates of registered clinical trials in macular degeneration. Retina. 2011 Feb;31(2):401-4.

We appreciate these additional reference list. We are aware of these publications and have added the Archer neonatal network example. 

Thank you again for the opportunity to revise and resubmit this manuscript.

---

## [Editor Report · Decision Letter 1]

30 Oct 2020

Publication Rates from Biomedical and Behavioral and Social Science R01s funded

by the National Institutes of Health

PONE-D-20-26044R1

Dear Dr. Riley,

We’re pleased to inform you that your manuscript has been judged scientifically suitable for publication and will be formally accepted for publication once it meets all outstanding technical requirements.

Kind regards,

Dr. Sakamuri V. Reddy

Academic Editor

PLOS ONE
---

## [Editor Report · Acceptance letter]

6 Nov 2020

PONE-D-20-26044R1 

Publication Rates from Biomedical and Behavioral and Social Science R01s funded
by the National Institutes of Health 

Dear Dr. Riley:

I'm pleased to inform you that your manuscript has been deemed suitable for publication in PLOS ONE. Congratulations! Your manuscript is now with our production department. 

Kind regards, 

on behalf of

Dr. Sakamuri V. Reddy 

Academic Editor

PLOS ONE